# *Helicobacter pylori*-Mediated Immunity and Signaling Transduction in Gastric Cancer

**DOI:** 10.3390/jcm9113699

**Published:** 2020-11-18

**Authors:** Nozomi Ito, Hironori Tsujimoto, Hideki Ueno, Qian Xie, Nariyoshi Shinomiya

**Affiliations:** 1Department of Surgery, National Defense Medical College, 3-2 Namiki, Tokorozawa, Saitama 359-8513, Japan; nozomi_i@ndmc.ac.jp (N.I.); tsujihi@ndmc.ac.jp (H.T.); ueno_surg1@ndmc.ac.jp (H.U.); 2Department of Biomedical Sciences, Quillen College of Medicine, East Tennessee State University, Johnson City, TN 37604, USA; xieq01@etsu.edu; 3Department of Integrative Physiology and Bio-Nano Medicine, National Defense Medical College, 3-2 Namiki, Tokorozawa, Saitama 359-8513, Japan; 4National Defense Medical College Research Institute, National Defense Medical College, 3-2 Namiki, Tokorozawa, Saitama 359-8513, Japan

**Keywords:** *Helicobacter pylori*, gastric cancer, signal transduction, LPS, MET, TLR4, CagA

## Abstract

*Helicobacter pylori* infection is a leading cause of gastric cancer, which is the second-most common cancer-related death in the world. The chronic inflammatory environment in the gastric mucosal epithelia during *H. pylori* infection stimulates intracellular signaling pathways, namely inflammatory signals, which may lead to the promotion and progression of cancer cells. We herein report two important signal transduction pathways, the LPS-TLR4 and CagA-MET pathways. Upon *H. pylori* stimulation, lipopolysaccharide (LPS) binds to toll-like receptor 4 (TLR4) mainly on macrophages and gastric epithelial cells. This induces an inflammatory response in the gastric epithelia to upregulate transcription factors, such as NF-κB, AP-1, and IRFs, all of which contribute to the initiation and progression of gastric cancer cells. Compared with other bacterial LPSs, *H. pylori* LPS has a unique function of inhibiting the mononuclear cell (MNC)-based production of IL-12 and IFN-γ. While this mechanism reduces the degree of inflammatory reaction of immune cells, it also promotes the survival of gastric cancer cells. The HGF/SF-MET signaling plays a major role in promoting cellular proliferation, motility, migration, survival, and angiogenesis, all of which are essential factors for cancer progression. *H. pylori* infection may facilitate MET downstream signaling in gastric cancer cells through its CagA protein via phosphorylation-dependent and/or phosphorylation-independent pathways. Other signaling pathways involved in *H. pylori* infection include EGFR, FAK, and Wnt/β-Catenin. These pathways function in the inflammatory process of gastric epithelial mucosa, as well as the progression of gastric cancer cells. Thus, *H. pylori* infection-mediated chronic inflammation plays an important role in the development and progression of gastric cancer.

## 1. Introduction

*Helicobacter pylori* is a microaerophilic bacterium that infects the gastric mucosal epithelium through colonization and triggers chronic inflammation [1]. Originally, it was discovered as a curved bacillus in the stomach of patients with gastritis and peptic ulceration by Marshall and Warren [2]. *H. pylori* produces proteases, vacuolating cytotoxin A (VacA), and certain phospholipases, all of which damage the gastric epithelial cells and disrupt tight junctions, thereby inducing inflammatory changes to gastric mucosa. *H. pylori* also produces urease that catalyzes the hydrolysis of urea to yield ammonia and carbonic acid, facilitating bacterial colonization by neutralizing gastric acid and providing materials for bacterial protein synthesis [3]. Chronic gastric inflammation caused by the products secreted by *H. pylori*, which stimulates transcription and synthesis of inflammatory cytokines, especially interleukin-1β (IL-1β) and tumor necrosis factor-α (TNF-α) [4], is considered to play an important role in the development and promotion of gastric cancer, and other organs may see an increased risk of cancer under such inflammatory conditions [5].

Although a persistent inflammatory condition is known to be key to *H. pylori*-induced gastric carcinogenesis, the molecular mechanisms underlying how *H. pylori* bacteria directly or indirectly interact with gastric epithelial cells, leading to gastric carcinogenesis, is unclear. In particular, how the inflammatory signals interact with intracellular pathways in gastric epithelial cells, ultimately leading to cell growth and differentiation, remains elusive.

We herein report the nature of *H. pylori*-induced inflammation and discuss the importance of bacterial-epithelial cell interaction in gastric cancer progression with a focus on the signal transduction pathways involved in gastric epithelial cells as well as inflammatory mediator cells, particularly the LPS-TLR4 and CagA-MET pathways.

## 2. *H. pylori* Infection and Immunity

*H. pylori*, like other pathogenic bacteria, produces and secretes various toxins outside of cells. Lipopolysaccharide (LPS) is a lipid A and polysaccharide complex comprising the outer membrane of the cell wall that is unique to Gram-negative bacteria. Unlike other secretory toxins, LPS remains on the outer membrane, the outermost layer of bacterial cell wall, and is thus also called an endotoxin. Once LPS binds to receptors on the target cell membrane, it impairs the cell function. LPS is the most potent microbial mediator contributing to the pathogenesis of sepsis and septic shock. When LPS enters the blood stream, it binds to and triggers mononuclear cells to produce pro-inflammatory mediators, including TNF-α and IL-1, which stimulate subsequent reactions in the neutrophil–endothelial cell adhesion and blood coagulation system to activate clotting and help generate micro-thrombi. Recent reports have indicated that chronic inflammation induced by LPS is involved in the progression of several diseases including obesity, type II diabetes, atherosclerosis, neuro-immune disorders, diverse metabolic diseases, and cancers [6,7,8,9,10,11].

LPS derived from *H. pylori* induces chronic inflammatory injury in gastric mucosa but has shown weaker endotoxic activity than that from Escherichia coli, likely due to the fact that lipid A (glycolipid terminal structure) of *H. pylori*-derived LPS does not have a typical β-1,6-diglucosamine skeleton, which is well-recognized in LPS derived from other Gram-negative bacteria, like *E. coli* [12,13]. Therefore, it is the *H. pylori*- but not *E. coli*-derived LPS that attenuates the cytotoxicity of mononuclear cells (MNCs) against gastric cancer cells. *H. pylori* LPS also downregulates the perforin production in CD56+ natural killer cells (NK cells) co-cultured with cancer cells, while *E. coli* LPS does not. In addition, it has also been reported that the milieu of *H. pylori* LPS is accompanied by the proliferation of regulatory IL-10-producing NK cells, which negatively regulates the cytotoxic activity of the gastric epithelia in the *H. pylori*-infected hosts. In contrast, NK cells in the milieu of *E. coli* LPS do not tend to produce IL-10 [14,15]. Thus, *H. pylori* LPS is considered to diminish the propagation and cytotoxic activity of NK cells that constitute the first line of anti-cancer immune defense, thereby leading to chronic infection of the gastric epithelia without severe cell damage [15]. In addition, the TNF-α mRNA expression is significantly lower in MNCs stimulated with *H. pylori* than in those stimulated with *E. coli* under the same conditions [16]. Although *H. pylori* LPS is reported to induce a high production of neutrophil-recruiting CXC chemokines similar to *E. coli* LPS, it is less potent for inducing pro-inflammatory cytokines in MNCs and a weak inducer of the CC chemokine RANTES (regulated on activation, normal T-cell expressed and secreted: RANTES) [17]. Importantly, while *E. coli* LPS is a potent inducer of TNF-α, IL-1β, IL-6, and IL-8, which function through the activation of nuclear factor-κB (NF-κB), *H. pylori* LPS inhibits most of these factors, including IL-1β, IL-6, and IL-8, and selectively upregulates the production of IL-18 and IL-12 [12]. (Figure 1). Specifically, *H. pylori* LPS binds to transmembrane Toll-like receptor 4 (TLR4) on the surface of MNCs and activates the TLR4- and toll/IL-1 receptor domain-containing adaptor inducing interferon-β (TRIF)-dependent pathways, which convert pro-IL-1β and pro-IL-18 to IL-1β and IL-18 [18,19]. Of note, IL-1β binds to its receptors on naïve T-cells and promotes Th1 and Th17 differentiation, leading to pro-inflammatory activities, whereas IL-18 counteracts IL-1β by preventing the Th17 response, Treg differentiation, and immune tolerance persistence, thereby balancing the control of infection [20,21,22]. The Th17 response is also reported to be driven by the *H. pylori*-secreted peptidyl prolyl *cis*, *trans*-isomerase, which promotes the pro-inflammatory low cytotoxic gastric tumor-infiltrating lymphocytes response, matrix degradation, and pro-angiogenic pathways, ultimately leading to the promotion of gastric cancer [23].

Thus, by selectively upregulating IL-18 but suppressing IL-1β, *H. pylori* LPS not only inhibits the immune inflammatory response but also promotes gastric cancer cells’ escape from immune surveillance, which facilitates gastric cancer initiation and progression.

Using an in vivo guinea pig model, the difference in the role of *H. pylori*-derived components such as glycine acid extract (GE), urease subunit A (UreA), CagA, and LPS has been discussed [24]. *H. pylori* infection promoted the infiltration of inflammatory cells in the gastric mucosa, which caused increased oxidative stress and cell apoptosis. Originally, the elimination of damaged cells by apoptosis has the effect of preventing deleterious inflammation and neoplasia. However, the upregulation of cell regeneration induced by GE, UreA, and CagA may be a risk factor for neoplasia. In contrast, *H. pylori* LPS had the effect of downregulating cell regeneration, which was shown to promote chronic inflammatory processes [24]. An in vitro study also showed that *H. pylori* LPS—but not CagA or other surface components of bacteria—inhibited cell migration and proliferation [25]. *H. pylori* LPS promoted chronic inflammation and inhibited the anti-bacterial response of immune cells, which may constitute the niche for neoplastic processes.

There are several reports on other negative immunomodulatory properties of *H. pylori* infection. The ability of mononuclear cells to present antigens to lymphocytes, lymphocyte proliferation, and phagocytic activity of the infiltrated macrophages were all adversely affected by *H. pylori* infection [26,27,28]. By showing these negative immunomodulatory mechanisms, *H. pylori* not only inhibits the inflammatory immune response and promotes their survival as well as long-term infection, but in turn also increases the risk of mutations of gastric epithelial cells. It helps the immune escape of cancer cells and contributes to the progression of cancer.

## 3. LPS-TLR Pathways in Gastric Cancer

Transmembrane TLRs are a class of signal transduction proteins referred to as pattern recognition receptors that can specifically recognize pathogen-associated molecular patterns (PAMPs), such as LPS, peptidoglycan, lipoteichoic acid, flagellin, double-stranded RNA; among those, TLR4 is identified as a receptor for LPS [29,30]. LPS binds to TLR4 to initiate the recruitment of adaptor molecules, such as MyD88, toll/IL-1 receptor domain-containing adaptor protein (TIRAP), TRIF, and TRIF-related adaptor molecule (TRAM). The further upregulation of transcription factors, such as NF-κB, activator protein-1 (AP-1), and interferon regulatory factors (IRFs), then follows [31] (Figure 2).

TLR4 is expressed in normal gastric mucosa and gastric epithelial cells in a highly polarized manner in apical and basolateral compartments [32]. The TLR4 expression is upregulated in gastric cancer tissues and confers LPS responsiveness to augment the activation of NF-κB and IL-8 promoter upon stimulation with *H. pylori* LPS [33]. When LPS binds to TLR4 in gastric epithelial cells, the activation of TLR4 signaling induces the synthesis of inflammatory mediators including TNF-α and IL-8, which are key factors promoting subsequent tumor development [34,35]. In MKN28 and MKN45 gastric cancer cells *H. pylori* LPS stimulation markedly upregulated the TLR4 expression and enhanced the IL-8 production, leading to an increase in cell proliferation [36].

In gastric cancers, studies have shown that TLR4 is not only overexpressed in gastric epithelia but also upregulated in monocytes/macrophages of superficial gastritis [32,37,38,39]. As mentioned above, *H. pylori* LPS may bind to and activate the TLR4 signaling pathway in MNCs and further suppress T cell-mediated cytotoxicity against gastric cancer cell proliferation (Figure 1). Our group has also reported a unique characteristic of the *H. pylori* LPS-TLR4 signaling axis in the progression of gastric cancers [40]. Furthermore, while *H. pylori* LPS as well as *E. coli* LPS was shown to augment the growth of gastric cancers, all of which expressed TLR4, the neutralization of TLR4 with specific antibodies almost completely abrogated *H. pylori*-induced gastric cancer cell proliferation. 

Taken together, these results support the importance of the LPS-TLR4 signaling pathway in gastric cancer proliferation and progression.

## 4. *H. pylori* Infection and CagA-MET Pathways

Cytotoxin-associated gene A (CagA) is an *H. pylori* protein known to be the primary cause of atrophic gastritis, which is a precancerous condition, which is known to cause peptic ulcerations [41]. During chronic infection of *H. pylori*, CagA is directly delivered into the target cells, such as gastric epithelial cells, via the type IV secretion system (T4SS), which is a secretory mechanism possessed by *H. pylori* [42,43,44]. Using a transgenic Drosophila model, Botham et al. reported that the CagA protein might function as a eukaryotic Gab1 adaptor protein that facilitates the activation of intracellular pathways related to cell proliferation and migration [44,45]. After delivery into host cells, CagA undergoes tyrosine phosphorylation on the Glu-Pro-Ile-Tyr-Ala (EPIYA) segments [46]. Phosphorylated CagA exerts various biological activities by interacting with host proteins, such as SHR2, CSK, CRK, and c-MET (Figure 3A) [47,48,49]. CagA-induced biological activities can be divided into three categories: scattering/hummingbird phenotype, disruption of tight junctions between cells, and activation of several transcription factors that control the cell proliferation, inflammation, and survival [50]. Thus, through the CagA-dependent pathways, *H. pylori* modifies the intracellular signals of target cells and promotes its infection and pathogenicity.

Given that *H. pylori*-derived LPS may stimulate gastric cancer cell proliferation via TLR4-IL-8 signaling, as mentioned above, *H. pylori* infection may also directly stimulate the proliferation and invasion of gastric cancer cells through the CagA-MET pathway [43,48]. Early studies have shown that *H. pylori* induced epithelial cell motility and MET phosphorylation similar to HGF/SF stimulation [51]. Mechanistically, activated MET recruits CagA as an adaptor protein and then induces CagA phosphorylation, leading to the activation of downstream phospholipase Cγ (PLCγ) and MAPK pathways (Figure 3B). Both PLCγ and MAPK inhibitors may attenuate *H. pylori*-induced cell motility. CagA may also activate MET signaling in a non-phosphorylated manner (Figure 3C). In such cases, a conserved motif in the C-terminal region of CagA known as the conserved repeat responsible for phosphorylation-independent activity (CRPIA) motif may interact with activated MET, leading to the activation of PI3K/Akt signaling for the proliferation of gastric cancer cells after *H. pylori* infection, which in turn contributes to the *H. pylori*-associated chronic gastric proliferative reaction and NF-kB signaling associated with the pro-inflammatory responses [48]. Thus, both the phosphorylated and non-phosphorylated forms of CagA can interact with MET and stimulate downstream signaling pathways in gastric cancer progression.

Furthermore, *H. pylori* infection-induced activation of MET may affect the immune cells that surround gastric cancer cells. It has been reported that *H. pylori* infection in gastric cancer cells increases the production of exosomes that contain the active form of MET [52]. Via an exosome-mediated transfer of activated MET, tumor-associated macrophages acquire the capacity to secrete the pro-inflammatory cytokine IL-1β and activate the Akt and MARK pathways, thereby inducing a tumorigenic effect on the gastric environment.

## 5. Other Signal Transduction Pathways

Signal transduction pathways other than TLR4 and c-MET also have cancer promoting activities to a substantial extent (Figure 4). Crosstalk between c-MET and epidermal growth factor receptor (EGFR) is involved in the progression of many types of cancers. In gastric cancers, *H. pylori* infection affects receptor molecules and induces transactivation of intracellular signals (Figure 4A). *H. pylori* mediates the activation of EGFR and stabilization on the cell surface by inhibiting its endocytosis and proteasomal degradation, which enhances the cell proliferation and survival in cooperation with c-MET downstream signals [53]. EGFR phosphorylation induced by *H. pylori* is known to activate the PI3K/Akt pathway [54]. In addition, it activates the small GTP-binding protein Ras, which in turn mediates ERK1/2 phosphorylation. Interestingly, this effect is much stronger in CagA+ strains of *H. pylori* bacteria than in the strains without CagA proteins [55]. It has been also reported that EGFR signal activation triggered by *H. pylori* enhanced IL-8 production, an important initiation factor of neoplastic transformation in gastric epithelial cells [56].

Focal adhesion kinase (FAK) plays an important role in the regulation of cell adhesion, spreading, motility, differentiation, and cell death [57]. Others have reported that, through binding to β-integrin of the cell membrane, *H. pylori* CagA may reduce FAK tyrosine phosphorylation in gastric epithelial cells [58]. This leads to impaired cell adhesion and increased motility, which may be involved in the development of gastric lesions associated with CagA-positive *H. pylori* infection (Figure 4B).

*H. pylori* is also known to stimulate the Wnt/β-Catenin signaling pathway which can cause uncontrolled cell growth and malignant transformation [59,60]. Studies have shown that *H. pylori* infection can function through Wnt/beta-catenin signaling to promote gastric cancer migration and invasion [60] The Wnt/β-catenin pathway is also involved in *H. pylori*-induced gastric cancer stem cell generation [61]. Experimentally, *H. pylori* upregulates Wnt10a and Wnt10b expression in gastric cancers, leading to Wnt/β-catenin pathway activation [62,63]. Finally, studies have shown that *H. pylori* induces rapid phosphorylation of the Wnt/β-catenin pathway co-receptor LRP6 independent of the CagA or VacA, a key toxin for *H. pylori* pathogenesis, leading to the accumulation of β-catenin in the nucleus [64] (Figure 4C).

## 6. Conclusions

*H. pylori* infection-associated chronic inflammation is an important factor in the development and subsequent progression of gastric cancer. The activation of the TLR4 pathway induced by *H. pylori* LPS markedly influences the immune response, which causes the attenuation of antitumor activity and induction of persistent infection and provides the environment to support tumor progression in gastric epithelial cells. Inflammation caused by *H. pylori* infection induces the overexpression of c-MET, and CagA released from *H. pylori* also enhances c-MET downstream signals. Recently, epigenetic disorder mechanisms have been indicated as causes of cancer associated with bacterial infection. The most relevant change is DNA methylation disorder, which allows the spread of *H. pylori* and sustained inflammatory processes [65]. To suppress the occurrence of *H. pylori* infection-associated gastric cancer, further research will be needed in order to develop new therapeutic strategies for controlling infection as well as optimizing the immune response to chronic inflammation.

## Figures and Tables

**Figure 1 jcm-09-03699-f001:**
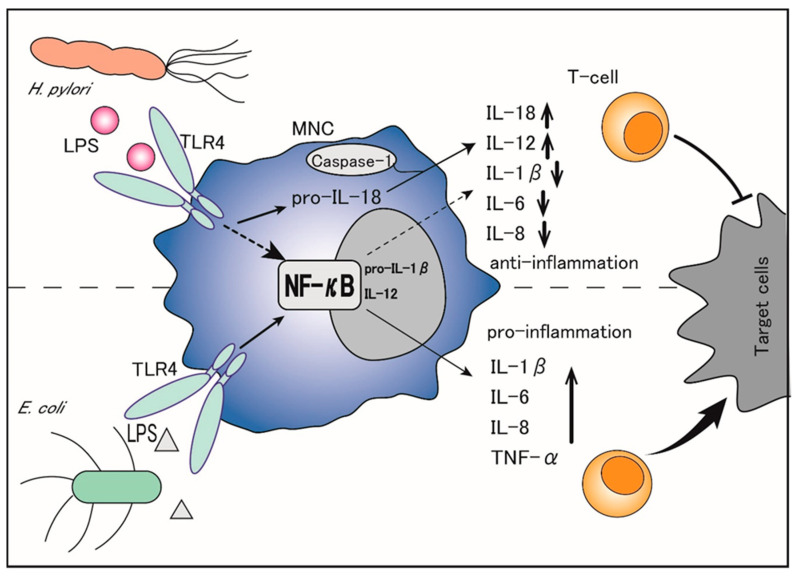
Difference in inflammatory reactions between *H. pylori* lipopolysaccharide (LPS) and *E. coli* LPS. *E. coli* LPS stimulates mononuclear cell (MNC) to upregulate the production of interleukin (IL)-1β, IL-6, IL-8, and tumor necrosis factor-α (TNF-α), whereas *H. pylori* LPS has an anti-inflammatory effect that selectively upregulates IL-18 and IL-12 but inhibits other cytokines, leading to the suppression of T cell surveillance, which promotes gastric cancer initiation and progression.

**Figure 2 jcm-09-03699-f002:**
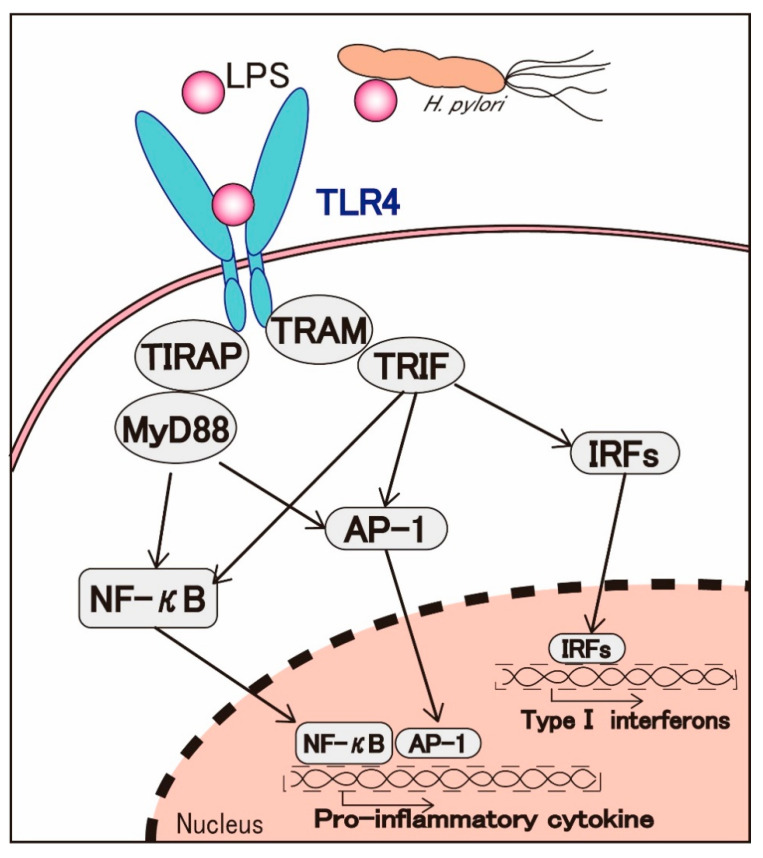
*H. pylori* LPS-mediated TLR4 signaling activation in gastric cancer. TLR4: toll-like receptor 4, TRAM: TRIF-related adaptor molecule, TIRAP: toll/interleukin-1 receptor (TIR) domain-containing adapter protein, TRIF: TIR domain-containing adapter inducing IFN-β, MyD88: myeloid differentiation factor 88, IRFs: interferon regulatory factors, NF-κB: nuclear factor-kappa B, AP-1: activator protein 1.

**Figure 3 jcm-09-03699-f003:**
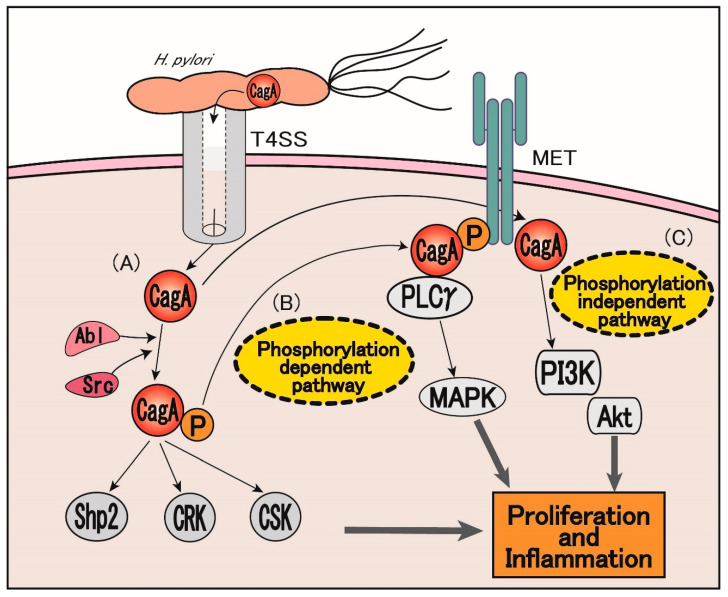
Induction of pro-inflammatory/pre-malignant response via CagA-MET signaling. (**A**) CagA mediated intracellular downstream signaling in gastric cancer cells. (**B**,**C**) CagA/MET signaling exerts its activity via phosphorylation-dependent (**B**) and phosphorylation-independent (**C**) pathways. T4SS: type IV secretion system, CagA: cytotoxin-associated gene A, PLCγ: phospholipase C gamma, PI3K: phosphatidylinositol-3 kinase, Akt: protein kinase B, PI3K: phosphatidylinositol-3 kinase, MAPK: mitogen-activated protein kinase, CSK: C-terminal Src kinase.

**Figure 4 jcm-09-03699-f004:**
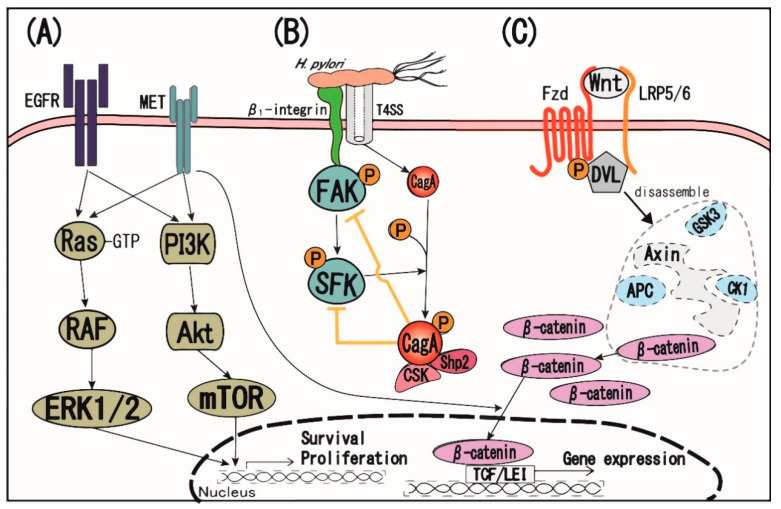
Other signal transduction pathways associated with *H. pylori* infection. (**A**) Crosstalk between the EGFR and MET pathways. (**B**) *H. pylori* directly binds to β1 integrins expressed on gastric epithelial cells. Activated β1 integrin stimulates focal adhesion kinase (FAK) and then upregulates Src family kinase (SFK) activity in the early phase of *H. pylori* infection. CagA delivered through the type IV secretion system into the host cytoplasm is rapidly phosphorylated by SFK. Tyrosine-phosphorylated CagA inactivates FAK and SFK by interacting with Shp2 and CSK in the late phase of *H. pylori* infection, eventually leading to impaired cell adhesion and increased motility. (**C**) Under baseline conditions, β-catenin, functioning as a protein transcriptional co-activator, is constantly phosphorylated by the Axin complex (also known as destructing complex). The interaction of Wnt glycoprotein ligands with Frizzled (Fzd) receptors and LRP5/6 co-receptors leads to the inhibition of the degradation of β-catenin by destructing complex. DVL: disheveled protein, CK1: casein kinase 1, GSK3: glycogen synthase kinase 3, APC: adenomatous polyposis coli, CSK: C-terminal Src kinase, mTOR: mammalian target of rapamycin.

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
