# Peer review of "Helicobacter pylori-Mediated Immunity and Signaling Transduction in Gastric Cancer"

_jcm, 2020, doi:10.3390/jcm9113699_

Round 1

Reviewer 1 Report

This is an interesting article, which treats about signalling pathways initiated by H. pylori compounds, such as lipopolysaccharide - LPS and cytotoxin associated gene A (XagA) protein, which may promote the development of gastric cancer: TLR4-LPS pathway and CagA-MET pathway. Both these pathways are involved in the development of inflammatory response on the level of gastric epithelial barrier and immunocompetent cells, mainly macrophages. The authors suggest an indirect relationship between H. pylori infection and the carcinogenic processes in the stomach through an increased inflammatory response.

However the Discussion can be conducted a bit more. 

Gonciarz et al., PLOS ONE, 2019, 14(8): e0220636, showed in Caviae porcellus model in vivo significant infiltration of the gastric mucosa by inflammatory cells in response to H. pylori, which was accompanied by elevated oxidative stress and cell apoptosis. These effects were more intense in an acute than chronic stage of infection. However, the increase of cell proliferation was more intense in chronic phase of infection. Furthermore,  in vitro study using primary gastric epithelial cells and fibroblasts of Caviae  showed that surface components of these bacteria (GE), CagA and LPS upregulated oxidative stress and apoptosis. Only H. pylori LPS inhibited cell migration and proliferation, which could affect the healing process of the gastric barrier (also Mnich et al., World J Gastroenterol, 2016; 22:75-36-7558). 

It is possible that LPS may induce deleterious effects due to induction of oxidative stress, and DNA damage. Theoretically, the intensification of cell apoptosis should protect against the development of cancer cells. However, cell loss due to apoptosis has been shown to increase gastric epithelial cell proliferation, which in turn increases the risk of mutations. In the case of H. pylori LPS, limiting the proregenerative capacity of cells may result in the chronic inflammation, which may constitute the niche for neoplastic processes. 

It has been shown that H. pylori components such as CagA or LPS, has negative immunomodulatory properties, which can facilitate the survival of bacteria and long lasting infection, thus increasing the risk for ulcer disease and cancer development. Particularly, LPS has been found to diminish propagation and cytotoxic activity of NK cells constituting the first line of anti-cancer immune defense. This is potentially related to the secretion of regulatory IL10 (Rudnicka et al., APMIS 2013, 121: 806-813; Rudnicka et al., Innate Immunity, 2015, 21:127-139).

In addition, the negative influence of H. pylori on the development of effective immune mechanisms towards these bacteria, e.g.by inhibition of the ability to present antigens of these bacteria to lymphocytes or inhibition their proliferation as well as due to inhibition of phagocytosis,  may create conditions favoring the neoplastic processes (Paziak-Domanska et al., Cell Immunol 2000, 202: 136-139; Schwartz et al., J Leukocyte Biol 2006, 79-1214-1225; Mnich et al., Acta Bioch Pol 2015, 62:641-650).

These are a few examples of how to expand the discussion with functional studies on LPS or CagA activity with potential relevance in the neoplastic process. They can enrich the discussion.

Author Response

To the Reviewer 1

Thank you very much for taking time to review our manuscript. We read your comments carefully and have re-written the paper according to your suggestions. Revised parts in the paper have been shown in red letters. The answer to each question is as follows:

Comment 1

Gonciarz et al., PLOS ONE, 2019, 14(8): e0220636, showed in Caviae porcellus model in vivo significant infiltration of the gastric mucosa by inflammatory cells in response to H. pylori, which was accompanied by elevated oxidative stress and cell apoptosis. These effects were more intense in an acute than chronic stage of infection. However, the increase of cell proliferation was more intense in chronic phase of infection. Furthermore, in vitro study using primary gastric epithelial cells and fibroblasts of Caviae showed that surface components of these bacteria (GE), CagA and LPS upregulated oxidative stress and apoptosis. Only H. pylori LPS inhibited cell migration and proliferation, which could affect the healing process of the gastric barrier (also Mnich et al., World J Gastroenterol, 2016; 22:75-36-7558).

Answer to Comment 1

According to your suggestion, we have added a paragraph to discuss the role of GE, UreA, CagA, and LPS in acute and chronic inflammatory settings associated with H. pylori infection. Therefore, the PLOS ONE and World J Gastroenterol papers have been cited as new references #24 and #25, respectively (Page 3, Line 109 to 119 in the revised manuscript).

Comment 2

It has been shown that H. pylori components such as CagA or LPS, has negative immunomodulatory properties, which can facilitate the survival of bacteria and long lasting infection, thus increasing the risk for ulcer disease and cancer development. Particularly, LPS has been found to diminish propagation and cytotoxic activity of NK cells constituting the first line of anti-cancer immune defense. This is potentially related to the secretion of regulatory IL10 (Rudnicka et al., APMIS 2013, 121: 806-813; Rudnicka et al., Innate Immunity, 2015, 21:127-139).

Answer to Comment 2

According to your suggestion, we have added the description about the role of H. pylori LPS and IL-10-producing NK cells (Page 2, Line 77 to 82 in the revised manuscript). Therefore, the APMIS and Innate Immunity papers have been cited as new references #14 and #15, respectively.

Comment 3

In addition, the negative influence of H. pylori on the development of effective immune mechanisms towards these bacteria, e.g.by inhibition of the ability to present antigens of these bacteria to lymphocytes or inhibition their proliferation as well as due to inhibition of phagocytosis,  may create conditions favoring the neoplastic processes (Paziak-Domanska et al., Cell Immunol 2000, 202: 136-139; Schwartz et al., J Leukocyte Biol 2006, 79-1214-1225; Mnich et al., Acta Bioch Pol 2015, 62:641-650).

Answer to Comment 3

According to your suggestion, we have added a new paragraph describing the importance of the other negative immunomodulatory properties of H. pylori infection (Page 3, Line 120 to 123 in the revised manuscript). Therefore, the Cell Immunol, J Leukocyte Biol, and Acta Bioch Pol papers have been cited as new references #26, #27, #28, respectively. Also brief summarized sentences have been added based on the content of the above three cited references (Page 4, Line 123 to 126 in the revised manuscript).

I hope the paper has been improved satisfactorily according to your suggestions. Again thank you very much for reviewing our paper.

Reviewer 2 Report

The manuscript is quite well written.
It represents a comprehensive review of the field.
It takes in account two important signal transduction pathways, the LPS-TLR4 and CagA-MET pathways.These pathways function in the inflammatory process of gastric epithelial mucosa, as well as the progression of gastric cancer cells. Thus, H. pylori infection-mediated chronic inflammation plays an important role in the development and progression of gastric cancer. It would be useful for the readers to include the discussion of PMID: 23054412.

Author Response

To the Reviewer 2

Thank you very much for taking time to review our manuscript. We read your comments carefully and have re-written the paper according to your suggestion. The answer to your question is as follows:

Your suggestion

To include the discussion of PMID:23054412; Helicobacter pylori secreted peptidyl cis, trans-isomerase drives Th17 inflammation in gastric adenocarcinoma

Answer to your suggestion

According to your suggestion, we have cited the PMID:23054412 paper as the reference #23, and the sentence has been added to the part where Th17 response is described. The Revised part in the paper has been shown in red letters (Page 3, lines 96 to 99).

I hope the paper has been improved satisfactorily according to your suggestion. Again thank you very much for reviewing our paper.

Reviewer 3 Report

The review of Ito N. et al. deals with the effect of Pylori in pathways leading to gastric cancer. First of all the title is misleading because it does not tell if the review deals with tumor initiation, tumor progression, mainteinance of the cancerous phenotype, or relationship between immunity and gastric cancer. Moreover, from the references the review appears outdated (the median age of the references is about ten years old) and a little autoreferential (for example reference 32). At least in one case, the references are wrong (for example, reference 23 refers to liver cancer not gastric). Also, some references are erroneously referred to (for example reference 12, where the authors state in the abstract that parts of pylori LPS are not strong inducers of IL-1 and IL-6 and IL-8; and not inhibitors as stated by the authors of the present review; while they show potent IL-18 and IL12 inducing activities). Sometimes plagiarism can be found (for example the sentence preceding reference 15). This review needs to be largely retyped.

Author Response

To the Reviewer 3

Thank you very much for taking time to review our manuscript. Also we are grateful to your insightful comments. We read your comments carefully and have re-written the paper according to your suggestions. The answer to each question is as follows:

Comment 1

First of all the title is misleading because it does not tell if the review deals with tumor initiation, tumor progression, mainteinance of the cancerous phenotype, or relationship between immunity and gastric cancer.

Answer to Comment 1

According to your suggestion, we have reconsidered the title of the paper. It has been changed to "Helicobacter pylori-mediated immunity and signaling transduction in gastric cancer". We have stated about the major focus of the paper in both the introduction (the last paragraph) and conclusion sections. It is not possible to cover all aspects, so we have focused those points. I hope the revised title explains the contents of the paper more accurately than the previous one.

Comment 2

from the references the review appears outdated (the median age of the references is about ten years old) and a little autoreferential (for example reference 32)

Answer to Comment 2

As you mentioned, some parts of the description refer to old literature, but they are related to the historical evidences that compose the premise of new findings. Nevertheless, your comment still has a point. So, we have added some new discussion parts, by citing recent literature (such as references #14, #15, #23-28) (Page 2, lines 77 to 123).

You mentioned there is a little self-citation, but we think the reference #32 (#40 in the revised manuscript) is an important and indispensable paper when talking about the LPS-TLR4 pathway in H. pylori-infected hosts. We appreciate your understanding.

Comment 3

At least in one case, the references are wrong (for example, reference 23 refers to liver cancer not gastric).

Answer to Comment 3

As you mentioned, yes, the reference #23 (#31 in the revised manuscript) is a review of TLR4 in liver cancer. But a half of the description is about the comprehensive TLR family, TLR4 ligands, and TLR4 signaling, and we quoted this reference for the purpose of discussing the TRL signaling in general. Therefore, we think it is no problem to cite this reference here. Again, we appreciate your understanding.

Comment 4

reference 12, where the authors state in the abstract that parts of pylori LPS are not strong inducers of IL-1 and IL-6 and IL-8; and not inhibitors as stated by the authors of the present review; while they show potent IL-18 and IL12 inducing activities

Answer to Comment 4

Thank you for your appropriate advice. We have corrected this part and rewritten the sentence (Page 2, lines 88 to 89). We have also corrected the related part of Figure 1.

Comment 5

Sometimes plagiarism can be found (for example the sentence preceding reference 15).”

Answer to Comment 5

Thank you for your kind criticism. Regarding the description related to the reference #15 (#17 in the revised manuscript), we did not intend to plagiarize the phrase. In order to dispel the misconception about this part, we have changed the wording (Page 2, lines 85 to 86).

I hope the paper has been improved satisfactorily according to your suggestions. Again thank you very much for reviewing our paper.

Round 2

Reviewer 3 Report

The paper has been improved according to my request and it is now suitable for publication